# Vitamin E in Cancer Treatment: A Review of Clinical Applications in Randomized Control Trials

**DOI:** 10.3390/nu14204329

**Published:** 2022-10-16

**Authors:** Jennifer Donnelly, Amanda Appathurai, Hui-Ling Yeoh, Kate Driscoll, Wasek Faisal

**Affiliations:** 1Department of Cardiothoracic Surgery, St.Vincent’s Hospital Melbourne, Fitzroy, VIC 3065, Australia; 2Nutrition Department, St.Vincent’s Private Hospital, Fitzroy, VIC 3065, Australia; 3Department of Infectious Diseases, The University of Melbourne, Parkville, VIC 3000, Australia; 4National Centre for Antimicrobial Stewardship, The Peter Doherty Institute for Infection and Immunity, Melbourne, VIC 3000, Australia; 5Department of Medical Oncology, Frankston Hospital, Peninsula Health, Frankston, VIC 3199, Australia; 6Department of Nutrition and Dietetics, Peninsula Health, Frankston, VIC 3199, Australia; 7Department of Medical Oncology, Ballarat Regional Integrated Cancer Centre (BRICC), Grampians Health, Ballarat Central, VIC 3350, Australia; 8School of Health, La Trobe University, Melbourne, VIC 3000, Australia

**Keywords:** Vitamin E, tocopherol, nutrition, cancer, malignancy, neoplasm, scoping review, randomized controlled trial

## Abstract

Vitamin E, along with other vitamins and micronutrients play a range of physiologic roles in the homeostasis of the body. Moreover, they also have postulated therapeutic roles that are often incompletely studied and understood. In this scoping review, we explored the recent randomized control trials (RCTs) of Vitamin E in the context of cancer, to investigate whether Vitamin E has a therapeutic role. We searched major bibliographic electronic databases to identify sixteen RCTs studying the role of Vitamin E in cancer management that have been published in the last ten years. These studies had different methodological qualities, including some that used Vitamin E in combination with other treatments. Furthermore, due to the heterogenous results, it is difficult to make a consensus statement on the effectiveness of Vitamin E in cancer therapeutics. In some cases, there were even suggestion of detriment with Vitamin E supplementation. Therefore, well designed, large, prospective RCTs are needed studying pure isoforms of Vitamin E to establish the safety and efficacy of this dietary supplement.

## 1. Introduction

Vitamin E comprises 8 natural molecules with similar chemical structure. They are divided in 2 subgroups (Tocopherols and Tocotrienols). Each group includes 4 isoforms (α,β,γ,δ). Vitamin E is an important dietary antioxidant, with sources including vegetable oils and nuts [1]. The antioxidant effect of Vitamin E has been credited with having a multitude of benefits, including anti-inflammatory, anticancer and neuroprotective effects. Beyond the regular dietary consumption, Vitamin E can also be regularly taken as a dietary supplement and is part of the billion-dollar dietary-supplement industry, with a projected estimated market capitalization of 200-billion dollars by 2025 [2]. 

α-Tocopherol has been traditionally considered as the predominant Vitamin E used in nutritional supplementation because of its higher tissue abundance and superior activity over other Vitamin E forms. γ-Tocopherol has been shown to effectively trap reactive nitrogen species and inhibit cyclooxygenase and therefore, having a stronger anti-inflammatory and anti-cancer effect than α-Tocopherol [3,4,5,6]. The robust cancer preventive activity of γ-Tocopherols has been demonstrated in animal models [3,4]. There have also been a number of studies examining the role of Vitamin E in the prevention of neurotoxicity from Platinum and Taxol-based chemotherapy regimens and alleviation of mucositis from radiotherapy toxicity, some of which will be reviewed in this paper. 

Gamma-tocotrienol is proven to be efficient in cancer therapy in both in vivo and in vitro studies [7,8,9]. In particular, γ-tocotrienol stimulates autophagy of cancer cells by a mechanism which inhibits the activation of mTOR and PI3 K pathways. Singh et al. [10] showed the induction of autophagy in human pancreatic cancer stem cells, while Tiwari et al. [11] demonstrated similar effects in human breast cancer tumor cells. Another mechanism described by Wali et al. [12] is the γ- tocotrienol triggered apoptosis in mammary malignant tumor cells through endoplasmic reticulum (ER) stress mediated apoptosis. Furthermore, in another article Tiwari et al. [11] proved the simultaneous induction of both ER stress mediated apoptosis and autophagy by γ-tocotrienol in human mammary malignant tumor cells. 

Being the major lipid-soluble, chain-breaking antioxidant in the body, Vitamin E protects the integrity of cell membranes by inhibiting lipid peroxidation and has a central role in maintaining neurological structure and function [13,14]. Vitamin E deficiency is a well-established cause of peripheral neuropathy hence a proven therapeutic agent in such cases [15,16]. There is however no current evidence in the literature regarding the effect of vitamin E in the treatment of idiopathic peripheral neuropathy. 

The finding that Vitamin E protected against Cisplatin-induced ototoxicity was demonstrated by a report showing that the guinea pigs injected with six daily doses of α-tocopherol had a significant amelioration of ototoxicity via the suppression of the increased production of reactive oxygen species and was further confirmed by Kalkanis et al. [17] that determined the effectiveness of vitamin E as otoprotectant in rats. Few studies have confirmed the neuroprotective role of Vitamin E against Platinum-induced neurotoxicity in humans, showing that patients taking Vitamin E had a lower incidence and severity of neurotoxicity with respect to patients receiving placebo [18,19]. Argyriou et al. [20] have also demonstrated a fifty-percent reduction in the incidence of peripheral neuropathy in patients undergoing Cisplatin chemotherapy who received 600 mg of oral Vitamin E per day, compared with those who received placebo. There have also been studies that have failed to demonstrate any meaningful clinical benefit of Vitamin E for the above-mentioned indications [21]. 

A major limitation of the data available on dietary supplements/micronutrients is that they are often based on cell-line or animal models, and human studies are mostly either unblinded, single-arm or small-scale, underpowered trials. This poses a significant challenge in obtaining high quality level one evidence to formulate a consensus recommendation. For the purpose of this review, we will be referring to Vitamin E as a single entity rather than its multiple isoforms, mainly due to the paucity of human studies investigating pure isoforms of Tocopherols and Tocotrienols. We will also review their effect on the pathophysiological processes involving cancer and therapeutics. This review will focus on the available randomized controlled trials of Vitamin E in humans in the context of cancer published in the last ten years, in order to understand the evidence-base in a contemporary light. 

## 2. Materials and Methods

The primary objective of this scoping review was to evaluate the contemporary evidence from randomized controlled trials of all isoforms of Vitamin E in human cancer treatment and pathophysiological processes involving cancer therapeutics. 

The methods used in this review are in line with the preferred reporting items for scoping reviews outlined in the PRISMA extension for scoping reviews (PRISMA-ScR) guidelines [22]. A systematic search strategy of electronic bibliographic databases (Ovid, PubMed and Cochrane CENTRAL) for published work in English language in the last ten years that included Vitamin E, cancer and treatment were conducted in September 2022 (example search strategy detailed in Appendix A). Studies were limited to those published with the relevant search terms, subjected to peer review in English language, involved in human adult subjects aged eighteen years and over, and dated from August 2012 to August 2022. Abstract and full text screening limited studies to randomized-controlled trials, relating to patients with diagnosis of solid organ cancer, and with Vitamin E as a treatment intervention. Studies were excluded if their primary outcome was cancer prevention, or if they were in relation to haematological malignancies.

After importing the reference list into the Covidence systematic review software, deduplication was carried out. Title and Abstracts were then reviewed by the first author (JD). Full text papers were then retrieved and reviewed by two authors (JD, HY) in parallel. Papers where reviewers disagreed on the rating of eligibility criteria were re-examined and discussed to reach consensus. Two review authors (JD, AA) independently extracted data including study identifiers, study design, population characteristics, interventions, intervention participation rates, control conditions, outcomes measures and follow-up data into a standardized extraction table. Data was extracted on the results and key outcomes defined by the study.

## 3. Results

The literature search identified a total of 1632 studies. After Covidence deduplication, a total of 739 studies were screened, with fifty-two studies remaining for full text screening. A total of sixteen studies were included in this review (Figure 1). 

All of the studies were single-center trials, with the country of origin in descending order being, Iran (six), India (three), Egypt (two), USA, Italy, UK, Korea and Brazil, each having one study. Participant numbers ranged from 18 to 140. The outcomes that the studies investigated were chemotherapy induced peripheral neuropathy [19,23,24,25,26,27], post treatment oral health including mucositis, xerostomia [28,29,30], nephrotoxicity, gastrointestinal adverse events, breast fibrosis [31] and antioxidant status in breast cancer [32], effects on prostate cancer [33], vaginal atrophy [34], and hearing loss [35] in cancer patients (see Table 1).

### 3.1. Peripheral Neuropathy

Six papers reported on the use of Vitamin E as a treatment for chemotherapy-induced peripheral neuropathy (CIPN). Among these, two studies examined oral Vitamin E (400 mg daily) in patients receiving Oxaliplatin-based chemotherapy regimens. Salehi et al. [19] found that colorectal cancer patients (n = 18) did not see a change in peripheral neuropathy scores from baseline to the end of sixth course of chemotherapy (mean difference (MD) intervention 6.37 ± 2.85 vs. control group 6.57 ± 2.94; *p* = 0.78). Afonseca et al. [23] recruited colorectal and gastric cancer patients (n = 32) and found an increase in peripheral neuropathy in the intervention group 83% vs. control group 68% using the Common Terminology Criteria for Adverse Events Version 3 (CTCAE V3). Diarrhoea was also more commonly observed in the intervention group (55.6% vs. 18.8%; *p* = 0.06).

Four studies investigated the effect of Vitamin E on CIPN in patients undergoing Taxol-based therapy. Anoushirvani et al. [24] recruited sixty-three patients with a variety of cancer types and reported a significant reduction in CIPN between intervention groups (Vitamin E 300 mg twice daily and Omega-3 640 mg three times per day) and placebo group (*p* = 0.001) with no significant difference between groups (*p* = 0.75). A larger study (n = 140) with solid and non-myeloid tumours reported no difference in the incidence of Grade 3/4 CIPN between the Vitamin E intervention group (400 mg twice daily) and the placebo group (25.7% in each arm, *p* = 1.0) as well as the time of onset (*p* = 0.24) [25]. However, the Vitamin E intervention group was associated with a significantly reduced median duration of CIPN (5 weeks) compared with the placebo group (12.5 weeks) (*p* = 0.01). Shamsaei et al. [27] recruited breast cancer patients receiving a Taxol-based chemotherapy regimen to determine if Vitamin E supplementation could inhibit the progression of CIPN. The intervention group (n = 35) received 400 IU of Vitamin E twice per day. A neurologist conducted electrophysiological assessments of the tibial, peroneal and sural nerves at baseline and 3 months post treatment. It found a decrease in sural nerve action in those receiving the intervention (*p* = 0.007) translating to a potential decrease in sensory nerve damage. A multi-arm study of lung, breast and ovarian cancer patients examined the effects of Vitamin E, methylcobalamin (Vitamin B12), glutamine and acetyl-L-carnitine (ALC) on the sensory, motor and pain symptoms of CIPN [26]. The Vitamin E treated arm (n = 21) (400 mg daily) found that sensory, motor and pain symptoms were relieved with comparable effect to methylcobalamin (*p* = 0.446), (*p* = 0.227). It provided better sensory, motor and pain relief than both glutamine (*p* < 0.001), respectively, and ALC (*p* = 0.002, *p* < 0.001, *p* < 0.001).

### 3.2. Oral Health

There were three studies that investigated the efficacy of Vitamin E in improving oral symptoms of mucositis, dysphagia, and xerostomia. Chung et al. [30] from Korea found that Vitamin E and C supplementation (100 IU Vitamin E plus 500 mg Vitamin C twice per day during radiotherapy) showed improvements in the xerostomia questionnaire score, as well as significantly better oral indices post-radiotherapy compared to the control group. Agha-Hosseini et al. [28] from Iran used a combination mouthwash to treat oral mucositis. They found the blend of Vitamin E, hyaluronic acid and triamcinolone acetonide was effective in treating oral mucositis and also reducing pain intensity. An Egyptian study found that the combined dose of pentoxifylline and Vitamin E had no effect on the incidence or onset of either oral mucositis or dysphagia, however, there was a reduction in the duration of both [29]. 

### 3.3. Miscellaneous Effects

There were seven other studies which investigated the effect of Vitamin E in participants with different cancers and outcome measures. Keshavarzi et al. [34] investigated the effect of Vitamin D and E vaginal suppositories on vaginal atrophy in women with breast cancer receiving Tamoxifen (n = 32). They found that there was a statistical significance in the Vaginal Maturation Index (*p* < 0.001) as well as reduction in vaginal pH (*p* < 0.001) and the symptoms of self-reported genitourinary atrophy also improved in the intervention group (*p* < 0.001).

A double-blinded, randomized control trial investigated the effects of Vitamin E (400 IU/daily) in the protection of Cisplatin-induced nephrotoxicity in participants undergoing Cisplatin chemotherapy for a range of different cancers (n = 49). Ashrafi et al. [36] found that there was a significant decrease in the blood levels of neutrophil gelatinase-associated lipocalin and serum creatinine in the Vitamin E group (*p* = 0.001). In another study which investigated the effect of Vitamin E (400 mg) as a protective factor against Cisplatin-induced ototoxicity in participants with a solid malignancy (n = 23) [35]. The control group experienced significant hearing loss compared to the intervention group (8000 HZ right ear: *p* = 0.04; left ear: *p* = 0.03) however, the audiogram showed no significant changes [35].

A British study [37] investigated the effects of Tocovid SupraBio 200 mg and pentoxifylline 400 mg orally twice daily for one year on those with gastrointestinal symptoms after radiotherapy for a pelvic malignancy (n = 62). The Tocovid SupraBio tablets contained D-α-Tocotrienol 61.52 mg, D-γ-Tocotrienol 112.80 mg, D-δ-Tocotrienol 25.68 mg, D-α-Tocopherol 91.60 IU. No significant improvements in gastrointestinal symptoms were found. 

Suhail et al. [32] investigated whether Vitamins C and E supplementation (Vitamin C = 500 mg, Vitamin E = 400 mg) provided protection against side effects of chemotherapy in patients with breast cancer (n = 80). They measured the degree of DNA damage by alkaline single cell gel electrophoresis in the peripheral lymphocytes. The authors found that DNA damage was significantly reduced in the intervention group compared to untreated breast-cancer patients (*p* < 0.001).

In another study, Jacobson et al. [31] investigated the effects of Vitamin E supplementation (400 mg) in combination with pentoxifylline on preventing radiation-induced fibrosis (n = 53). They found that there was a trend towards a difference between tissue compliance meter measurements (*p* = 0.05). Goodin et al. [33] assessed the effect of Vitamin E supplementation for seven or fourteen days prior to prostatectomy on tocopherols levels and their metabolites in prostate cancer (n = 59) and found that side-chain degradation metabolites were significantly increased after Vitamin E supplementation (*p* < 0.05). 

## 4. Discussion

Vitamin E is a fat-soluble vitamin within the Tocols family. The Tocol family include four tocopherols and four tocotrienols (α, β, γ and δ). Tocols have the role of maintaining cell membrane fluidity in cell membranes. They protect cells against adverse effects of free radicals by acting as biological antioxidants. They protect lysis of red blood cells after irradiation. Vitamin E has been documented as potentially having multiple effects mediated by endothelial nitric oxide synthase including vasculoprotective, anti-inflammatory and anti-fibrotic effects [6,38]. 

The aim of this scoping review was to review the human RCTs investigating the effect of Vitamin E on oncological treatment-related side-effects. The most studied outcome was chemotherapy-induced neurotoxicity, with results that can be described as contradictory at best. Some of the studies showed a benefit while others failed to do so, and even showed detriment. Moreover, some of these studies did not use Vitamin E as a single intervention, but rather, as part of a combination of other agents, making it very challenging, if not impossible, to draw any profound conclusion from their findings. Similarly, on the question of maintaining oral health, the two studies showed different outcomes, but again, Vitamin E was used as part of a combination regimen, thereby lending itself to the same criticism. Overall, all the studies were single-center studies, with a small sample size and heterogenous design, posing significant challenges to draw a meaningful conclusion. 

In the last twenty years, the body of literature has established a thorough understanding of Vitamin E metabolism and the biological roles of its metabolites [6]. Presently, there is extensive literature on the use of vitamin E for cancer treatment in vitro and animal studies [39]. Unfortunately, our study concludes that such studies are limited in in vivo environments involving patients undergoing treatment for cancer. Greater research focus appears to be present in the efficacy of Vitamin E in cancer prevention in humans rather than in the context of cancer treatment [40,41]. There is also a growing body of work on the role of Vitamin E in nanomedicine. Given the role of Vitamin E as an antioxidant, its influence on connective tissue growth factor, gene expression and well documented benefits in wound healing, research in local administration of Vitamin E might prove more favorable than systematic administrations of Vitamin E in cancer treatment in humans [42]. 

Some studies have also suggested that Vitamin E in its various forms may have a role to play in mediating the effects of the Systemic Inflammation Index (SII) and therefore potentially increases the chances of long-term survival for people undergoing treatment for cancer [34,35]. The SII has been used in trials investigating the effects of Vitamin D to objectively measure its inflammatory effects in humans [36]. As inflammatory biomarkers and cytokines are routinely measured throughout the course of a patient’s disease, there is a potential role for SII to be used in future Vitamin E clinical trials [37].

This review has some limitations. Firstly, only RCTs in English published in the last ten years were included, which may result in selection bias. Additionally, the heterogeneity of the studies, patient characteristics, different malignancies and treatment regimens used resulted in poor comparability between trials. Lastly, although it has been established that different isoforms of Vitamin E have varying pharmacokinetic effects, no clear distinction could be made between the isoforms of Vitamin E that was administered in each study, and therefore, all results were attributed to Vitamin E as a single entity. 

Although our aim was to review studies of the role of Vitamin E in cancer treatment, it has become apparent all existing RCTs involved the adjuvant role of Vitamin E in prevention and improvement of symptoms due to secondary effects of other therapeutic agents. Despite the above-mentioned limitations of the present RCTs, there appears to be conflicting results between in vitro studies [9,10,11,12,17,18,19,20,21,43] showing clear beneficial roles of Vitamin E in affecting survival and proliferation of cancer cells and recent RCTs failing to investigate such effects or finding minimal to no benefits in management of therapy related detrimental secondary effects. The results of the available RCTs do not allow for a meaningful conclusion on the beneficial role of Vitamin E in the management of the chemotherapy-related side effects. While it is possible that Vitamin E may mitigate the severity of CIPN and help to maintain oral health, these studies were varied in their methodology and had significant limitations, including small cohort numbers and heterogenous interventional doses. This precludes our ability to draw a strong conclusion about the efficacy of Vitamin E as a protective nutritional supplement. Further high-quality randomized controlled trials in humans are required. 

## Figures and Tables

**Figure 1 nutrients-14-04329-f001:**
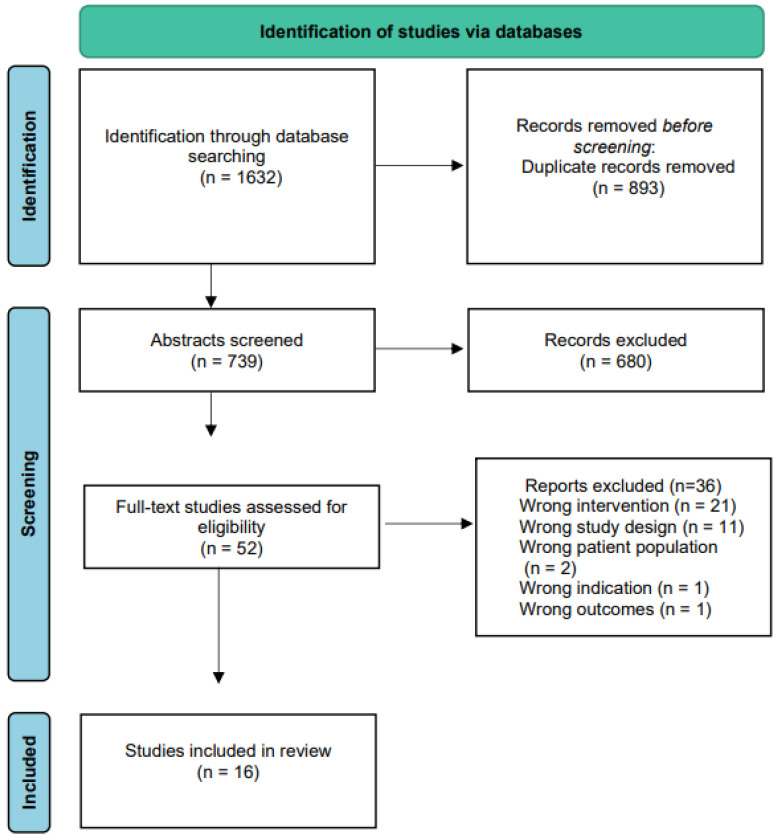
Consort diagram.

**Table 1 nutrients-14-04329-t001:** Characteristics of included RCTs.

First Author, Country, Year	Study Design	Delivery Method	Participants	Interventions	Other	Intervention Participation Rate	Control	Outcome Measures	Follow-Up after End of Active Intervention
Keshavarzi et al. [34], Iran 2019	Triple blind RCT	Vaginal suppository inserted with applicator, every day before bedtime	Breast cancer patients receiving Tamoxifen	Vitamin D (0.025 mg) suppository. Vitamin E (1 mg) suppository.	Placebo suppository	N = 64 female participants were randomized to each intervention group.	n = 32 females	Reduction in the vaginal pH, increase in vaginal maturation index, and improvements in the subjective symptoms of vaginal atrophy	8 weeks of using the intervention daily, with visits in the first, second and fourth week, then eighth week.
Mondal et al. [26], India 2014	Single site, prospective, open-label, multi-arm, randomized control study	Oral dose	Patients with lung, breast, or ovarian cancers, on paclitaxel 175 mg/m^2^ intravenous as 1st or 2nd line drug.	Arm A (vitamin E 400 mg once daily day 1 of the cycle to 1 month after completion of clinical trial	Arm B (ALC 250 mg OD from day 1 to day 7 in each cycle of chemotherapy (CTx)); Arm C (glutamine 10 mg TDS from day 2 to day 5 in each cycle), and; Arm D (methylcobalamine 500 μg TDS from day 1 of the first cycle to 1 month after completion of trial.	160 recruited, n = 21 in the intervention	22, 24 and 23 in other arms of the study	Clinical assessments for motor, sensory and pain components of peripheral neuropathy were done using the Common Terminology Criteria for Adverse Events (CTCAE) v4.02, 15 September 2009 criteria.	Monthly assessments post treatment. In patients where muscle weakness was suspected, neurological check-ups at a neurological tertiary care center.
Salehi et al. [19], Iran 2015	Open label RCT	Oral dose	Colorectal cancer patients to receive Oxaliplatin based CTx enrolled to evaluate Vitamin E on prevention of Oxaliplatin induced peripheral neuropathy	Vitamin E, 400 mg daily	No placebo for control group	Total 65 participants Intervention n = 32	Control n = 33	Peripheral neuropathy score changes (after − before) between two groups, after sixth course of the Oxaliplatin. Patient reported symptom experience diary questionnaire	Follow up for 6 courses of CTx
Sayed et al. [29], Egypt 2019	Open label RCT	Oral dose	Adults with squamous cell carcinoma of the head and neck eligible for treatment with Radiotherapy (RTx).	30 patients had 30–35 fractions of RTx to a total of 60–70 Gray (Gy) over 6–7 weeks. Additionally, Pentoxifylline 400 mg oral tablets twice daily and vitamin E 1000 mg oral soft gelatin capsules once daily. Twenty patients had concurrent Cisplatin 100 mg/m^2^ every 21 days.	30 patients who had 30–35 fractions of 3D conformal RTx to a total of 60–70 Gy over 6–7 weeks. Nineteen of those patients received concurrent cisplatin 100 mg/m^2^ every 21 days.	Total randomized n = 60. Intervention n = 22	Control n = 25	Patients interviewed to assess oral mucositis and/or dysphagia. Graded using CTCAE v4.03 Functional oral intake scale (FOIS) score, Arabic version of the Euro Quality of Life (QOL)-5D-3L at baseline and after 90-day follow-up. EQ-5D-3L converted into EQ-5D Index scores, used with patients’ Visual Analogue Scale (VAS) for patients’ QOL	90 day follow up
Shamsaei et al. [27], Iran, 2017	Double-blind RCT	Oral dose	Patients with breast cancer, and on a Taxol-based CTx regimen.	Vitamin E ‘feeding’, a daily dose of 400 IU	Placebo	Total randomized n = 70 Intervention n = 35	n = 35	Clinical and electrophysiological evaluation by a neurologist. Standard neurophysiological test performed unilaterally (on the right side). Electrophysiological assessment of tibial and peroneal nerves motor conduction. Amplitude of sensory action potential measured to evaluate nerve sensory conduction	3 months
Suhail et al. [32], India, 2010	Prospective RCT	Oral dose	Women between 35 and 65 years, with breast cancer. Only patients with stage II of TNM classification, with no previous treatment	Vitamin E, Tocopheryl Acetate 400 mg as Evion 400 IU gelatinous capsule and Vitamin C as Limcee 500 mg tablet once a day during CTx and for 3 weeks after CTx cessation	CTx alone, or untreated groups	Two groups: CTx alone (n = 20) and CTx + Vitamins C and E (n = 20)	Two groups: Control (n = 40) and Untreated (n = 40)	Plasma measured to check antioxidant enzymes, superoxide dismutase (SOD), catalase (CAT), glutathione-S-transferase (GST) and glutathione reductase (GR) and the levels of malondialdehyde (MDA) and reduced glutathione (GSH). DNA damage assessed in the peripheral lymphocytes	5 months
Villani et al. [35], Italy 2016	Phase III, randomized, placebo controlled trial.	Oral dose	Patients with solid malignancies and good functional status (Karnofsky Performance Status 70–100) candidates to receive Cisplatin CTx were enrolled in this study.	Oral vitamin E supplementation at the dosage of 400 mg per day during CDDP (Cisplatin) treatment	Placebo-composed of rice powder and were identical to vitamin E tablets	n = 54 assigned, n = 13 included in analysis. n = 41 excluded	n = 54 assigned, n = 10 included in analysis. n = 44 excluded	At baseline (T0), after 1 month (T1), 2 months (T2), and 3 months (T3), all patients evaluated with audiograms and evoked brainstem responses. By a trained operator, blinded to patients’ clinical and treatment status.	3 months
Anoushirvani et al. [24], Iran, 2018	Randomized controlled trial.	Oral dose	Aged 30–70 years who received Taxol CTx.	Arm A: 640 mg Omega-3 three times per day Arm B: 300 mg Vitamin E two times per day All interventions were for 3 months after onset of Taxol	Placebo	Total randomized n = 63 Arm A n = 15 Arm B n = 16	n = 15 men n = 16 women	Before CTx, 1 month and 3 months, patients were reviewed by a neurologist for electrophysiological evaluation.	1 and 3 months during intervention
Ashrafi et al. [36], Iran, 2020	Prospective, double-blinded, placebo controlled RCT.	Oral dose	Cisplatin-based CTx patients	Vitamin E supplementation (400 IU/daily) (pearl tablet) n = 26 Commenced 1 day before the initiation of CTx and continue along CTx cycles until three weeks post treatment.	Placebo	n = 64 8 lost to follow-up Intervention n = 26	n = 25	Serum creatinine, glomerular filtration rate (GFR), and neutrophil gelatinase-associated lipocalin measured prior to each CTx cycle and one month post cycle	One month post end of last CTx cycle.
Heiba et al. [25], Egypt, 2021	Prospective phase II, open-label RCT	Oral dose	Patients receiving taxane-based CTx in Ain Shams University Hospitals	Vitamin E 400 mg twice daily starting with CTx and for 1 month after its completion. n = 70	Same CTx without vitamin E prophylaxis.	n = 140 randomized n = 70 Intervention	n = 70	Patients assessed for peripheral neuropathy using CTCAE v 5.0 criteria. Peripheral neuropathy assessment completed before each CT cycle, and 1 and 3 months post CTx completion.	3 months post completion of CTx
Andreyev et al. [37], UK, 2022	Randomized controlled trial.	Oral dose	Patients over 18 years, radical RTx for a malignant pelvic neoplasm of the prostate, testis, bladder, uterine cervix, uterus, vagina, vulva, rectum (treated without surgery), anal canal or ovary.	ToCOVID SupraBio & Pentoxifylline (n = 40). n = 38 received intervention.	Placebo	n = 62 randomized. n = 38 intervention	n = 21	Baseline 7-day food dairy and interview with a Dietitian to ensure patients maintain a stable fat intake. At 3, 6, 9, 12 and 24 months patients completed clinical assessment, CTCAE assessment and QOL questionnaires.	Patients were followed up 24 months post randomisation
Chung et al. [30], Korea, 2016	Prospective, double-blind RCT	Oral dose	Aged over 18 years, with biopsy-proven head and neck cancer and treated with >4000-cGy intensity-modulated RTx.	Antioxidant supplements (100 IU vitamin E + 500 mg of vitamin C) twice per day during RTx.	Identical pill placebo	n = 45 n = 25 intervention	n = 20	Pre-RTx, one and six months post-RTx, patient reported xerostomia questionnaires, observer-rated xerostomia score, and salivary scintigraphy to compare xerostomia severity between the 2 groups.	One and six months post RTx
Afonseca et al. [23], Brazil, 2013	Phase III RCT	Oral dose	Gastric and colorectal cancer patients commencing oxaliplatin treatment	Vitamin E	Placebo	n = 34 Intervention n = 18	n = 16	CTCAE version 3, and specific gradation scales for Oxaliplatin induced peripheral neuropathy	Nil
Jacobson et al. [31], US, 2013	Prospective, randomized controlled trial.	Oral dose	Patients with breast cancer treated with breast conservation or after mastectomy, without metastatic disease. Patients had a 3D treatment plan based on a planning CT scan.	Oral PTX 400 mg 3 times per day and oral vitamin E 400IU daily for 6 months after radiation	Standard care	n = 53 randomized n = 26 in the intervention group	n = 27	Tissue compliance measurements at 18 months to compare tissue compliance in the irradiated and untreated breast/chest wall in both groups. Measurements were obtained at 2 mirror image sites and the difference scored.	18 months, then minimum of 2 years for local recurrence, disease-free survival
Agha-Hosseini et al. [28], Iran, 2021	Triple blinded RCT	Oral dose	Patients with any type of head and neck malignancy undergoing RTx on an OP basis, referred to the university’s cancer institute.	0.1% triamcinolone, 0.2% vitamin E and 0.2% hyaluronic acid. n = 29	0.1% triamcinolone mouthwash alone n = 30	n = 59 randomized Intervention n = 29	n = 30	Oral mucosa analysed for mucositis each week for four weeks. Pain intensity was assessed using a numerical pain intensity scale	4 weeks
Goodin et al. [33], US, 2021	Pilot Phase 0 RCT	Oral dose	Men with prostate cancer undergoing radical prostatectomy	Arm A: Daily oral high y-T-rich vitamin E supplementation for 1 week prior to prostatectomy Arm B: Daily oral high y-T-rich vitamin E supplementation for 2 weeks prior to prostatectomy	Arm C: No supplementation	n = 59 randomized Arm A: n = 16 Arm B: n = 19 received allocated intervention	n = 21	Blood and urine samples collected before supplementation and on the day of surgery, along with prostate tissue, for analysis of tocopherols and their metabolites.	Nil

## Data Availability

Not applicable.

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
