# Peer review of "Vitamin E in Cancer Treatment: A Review of Clinical Applications in Randomized Control Trials"

_nutrients, 2022, doi:10.3390/nu14204329_

Round 1
Reviewer 1 Report
The authors that summarized the updating information on Vitamin E. They are explored Vitamin E of the recent randomized control trials (RCTs) in cancer, to investigate whether Vitamin E has a therapeutic role.
It is important and interesting. Some comments is as following:
1. In the review should provide a cartoon scheme to summary the role of Vitamin E is how to involve in cancer controlling for more ease to catch the points.
2. The authors should listed a Table for how many cancer using or treated by Vitamin E, and possible mechanisms in detail.
3. The role of Vitamin E should mention whether play some important functions on cell death control or autophagy induction that they can address its role in clinical cased.
3. In Table 1 is not ease to catch the point should be more precise.
Reviewer 2 Report
The authors reviewed extensive literature to identify sixteen single-centered recent randomized clinical trials over the 2012-20 period that interrogated the effect of Vitamin E on oncological treatment-related side effects, largely related to chemotherapy and radiation therapy and most frequently chemotherapy-induced neurotoxicity. The well-presented results are identified as being contradictory “at best” and did not yield meaningful data showing a beneficial role for Vitamin E in the management of chemotherapy-related side effects. The many cofounders compromising the interpretation of the previous studies are well recognized and put forth. The studies are comprehensive, well designed, interesting, and present novel information of particular interest to the oncology and toxicological literature.
However, there is the issue that many recent studies and therapeutic strategies strongly support the concept that reactive oxygen species have an anti-tumorigenic role once the tumor is initiated and that “antioxidants” act in a pro-tumorigenic manner (e.g. Wiel et al, Cell 178: 330, 2019 and included references). The most important shortcoming of the paper as submitted is that it gives the reader the impression that Vitamin E could be innocuous to the tumor itself.
Comments
- The belief that antioxidant supplements protect against all phases of cancer including initiated cancers persists in society. The authors need to (i) be clear that their study is focused on oncological-induced treatment-related side effects as clearly put forth in lines 208-210. This description of the study needs to come across in Title and Introduction more clearly. (ii) The fact that existing literature suggests that the Vitamin E given for side effects may be detrimental to treatment strategies of the tumor itself needs to be recognized.
- It might be mentioned more strongly that local administrations of Vitamin E might be more helpful than systematic administrations (e.g., oral health).
- References would be helpful for lines 41 and 226.
- Are there data indicating that Vitamin E is useful (or not) in idiopathic forms of peripheral neuropathy? And if so, it would be helpful to mention.
